# COVID-19 in Children and Vitamin D

**DOI:** 10.3390/ijms252212205

**Published:** 2024-11-14

**Authors:** Teodoro Durá-Travé, Fidel Gallinas-Victoriano

**Affiliations:** 1Department of Pediatrics, School of Medicine, University of Navarra, 31008 Pamplona, Spain; 2Navarrabiomed (Biomedical Research Center), 31008 Pamplona, Spain; fivictoriano@hotmail.com; 3Department of Pediatrics, Navarra Hospital Universitary, 31008 Pamplona, Spain

**Keywords:** COVID-19, MIS-C, SARS-CoV-2 infection, vitamin D, vitamin D deficiency, vitamin D insufficiency, vitamin D supplementation

## Abstract

In December 2019, the so-called “coronavirus disease 2019” (COVID-19) began. This disease is characterized by heterogeneous clinical manifestations, ranging from an asymptomatic process to life-threatening conditions associated with a “cytokine storm”. This article (narrative review) summarizes the epidemiologic characteristics and clinical manifestations of COVID-19 and multi-system inflammatory syndrome in children (MIS-C). The effect of the pandemic confinement on vitamin D status and the hypotheses proposed to explain the age-related difference in the severity of COVID-19 are discussed. The role of vitamin D as a critical regulator of both innate and adaptive immune responses and the COVID-19 cytokine storm is analyzed. Vitamin D and its links to both COVID-19 (low levels of vitamin D appear to worsen COVID-19 outcomes) and the cytokine storm (anti-inflammatory activity) are detailed. Finally, the efficacy of vitamin D supplementation in COVID-19 is evaluated, but the evidence supporting vitamin D supplementation as an adjuvant treatment for COVID-19 remains uncertain.

## 1. Introduction

In December 2019, there was an outbreak of a new infectious disease in Wuhan, the capital of the Hubei province (China) caused by a new type of coronavirus: severe acute respiratory syndrome coronavirus 2 (SARS-CoV-2), and later named “coronavirus disease-2019” (COVID-19) [1]. In February 2020, the Chinese Center for Disease Control and Prevention (China CDC) provided important information to the international community about the SARS-CoV-2 infection: “The COVID-19 epidemic had spread very rapidly and it took only 30 days (from 31 December 2019 to 11 February 2020) to spread from Hubei Province to the rest of mainland China” [2]. On 11 March 2020, the World Health Organization (WHO) declared a public health emergency of international concern, but the virus spread rapidly causing a global pandemic. COVID-19 exhibits highly heterogeneous clinical manifestations, ranging from an asymptomatic or paucisymptomatic process to a life-threatening condition characterized by features of interstitial pneumonia, acute respiratory distress syndrome (ARDS), and severe multi-organ failure [3]. It is transmitted from human-to-human in a variety of ways (direct contact or indirect contact (fomites), small airborne droplets or aerosols) but primarily by large, inhaled droplets from coughing or sneezing [4,5]. Although the first data available on COVID-19 suggests possible animal-to-human transmission, probably from bats, epidemiological data have increasingly shown that the virus is transmitted from human-to-human through droplets or direct contact [6,7].

In addition, the introduction of vaccines against SARS-CoV-2 changed the course of the pandemic. In fact, the recent development of vaccines was considered an effective measure to save lives and minimize the impact on health, social systems and the global economy [8]. SARS-CoV-2 genome mutations are known to influence the efficacy of the immune response induced by vaccination. Since the beginning of the COVID-19 pandemic, numerous SARS-CoV-2 mutations have been identified, and regular viral genome sequencing helps to detect new genetic variants circulating in communities. An updated version of the SARS-CoV-2 phylogenetic tree is shared on the GISAID (Global Initiative on Sharing Avian Influenza Data) platform. A variant is recognized by the WHO as a variant of concern (VOC) or variant of interest (VOI) [9].

The China CDC reported the first description of the demographic characteristics of the COVID-19 outbreak of the novel coronavirus SARS-CoV-2 in February 2020. Of the total 44,672 confirmed cases of COVID-19 in China, only 2% were patients aged 0–19 years [2]. Italian data, published on March 2020, reported that only 1.2% of 22,512 Italian cases of COVID-19 were children [10]. Of the total 44,672 COVID-19 cases reported in the United States as of March 2020, only 5% of cases occurred in patients aged 0–19 years [11]. According to a multicenter retrospective observational cohort study of 8886 pediatric patients (0 to 18 years) diagnosed with COVID-19, the most common age group was over 10 years (57.0%). The proportion of disease was 8.4% in children less than 1 year of age, 20.7% in the 1–6 age group, and 13.9% in the 6–10 age group [12]. However, some children may experience a post-infectious inflammatory process temporally related to COVID-19 named “multisystem inflammatory syndrome” (MIS-C) [13,14,15].

From the early stages of the pandemic, it was observed that there was a significant difference in the severity of COVID-19 in different age groups: elderly patients infected with SARS-CoV-2 are at high risk for ARDS, complications, and death, while children with COVID-19 showed milder cases and a better prognosis than adults, and death was relatively rare [16,17,18,19,20,21,22]. The results found in a recent meta-analysis, after analyzing a large population group of patients (611,583 subjects) with COVID-19 from several national and regional registries (China, Italy, Spain, the United Kingdom and New York State), confirm that age has a determining effect on the mortality of COVID-19 patients. According to age, mortality in patients younger than 50 years was 0.3–1.1% and increased exponentially with age: 3.0% (50 to 59 years), 9.5% (60 to 69 years), 22.8% (70 to 79 years) and 29.6% (over 80 years) [23]. In some patients, COVID-19 infection is often accompanied by an aggressive inflammatory response with the release of large amounts of pro-inflammatory cytokines, a so-called “cytokine storm”. In addition, several studies suggested that a cytokine storm correlates with lung injury, multi-organ failure, and poor prognosis in severe COVID-19 [24,25,26]. In addition, recent clinical data [27] have shown that patients with COVID-19 have significantly elevated levels of matrix metalloproteinase (MMPs), which are known to play a role in tissue remodeling and immune responses. In other words, MMPs, together with the cytokine storm, may play an important role in the immunopathogenesis of COVID-19.

At present, vitamin D deficiency is a global health problem that affects all ages and races. The important discovery that vitamin D can act on various cell types through the vitamin D receptor (VDR) opened a wide field for investigation of its role in human health. As a result, VDRs (nuclear transcription factors) have been detected in almost all immune cells, and their activation would regulate the expression of many genes involved in both the innate and adaptive immune systems. Therefore, vitamin D status in COVID-19 could play an important immunomodulatory role by regulating the pathophysiological manifestations of the cytokine storm (anti-inflammatory agent). However, it remains to be seen whether vitamin D supplementation is associated with a potentially favorable outcome of COVID-19 [28,29,30].

The objective of this review is to provide a comprehensive literature review (narrative review) of (a) recent data on vitamin D and its association with COVID-19 and a cytokine storm and (b) clinical evidence between COVID-19 and/or MIS-C and vitamin D, and finally, (c) evaluate the efficacy of vitamin D supplementation in pediatric patients with COVID-19. This review is based on an electronic search of the literature performed by two independent researchers in the PubMed database of the U.S. National Library of Medicine published between January 2020 and August 2024. The following specific keywords (Medical Subject Headings) were used alone or in combination for the search: children, COVID-19, MIS-C, SARS-CoV-2 infection, vitamin D or vitamin D deficiency/insufficiency and vitamin D supplementation.

## 2. The COVID-19 Cytokine Storm

The normal response to a viral infection requires activation of the inflammatory pathways of the immune system. The inflammatory response to the invading virus results in the activation of transcription factors that induce the expression of genes encoding various pro-inflammatory cytokines. Cytokines are produced by different immune cells from both the innate immune system (macrophages, dendritic cells, natural killer cells) and the adaptive immune system (T and B-lymphocytes). The clinical findings are mainly attributed to the action of the pro-inflammatory: interleukin (IL)-1, IL-6, IL-8, tumor necrosis factor-alfa (TNF-α) and interferon-gamma (IFN-γ). This increase in cytokines results in the influx of several immune cells (macrophages, neutrophils, T cells, etc.) from the circulation to the site of infection, with destructive effects on human tissues (damage of vascular barrier and hemodynamic instability, diffuse alveolar damage, multi-organ failure, etc.) and, if left untreated, death [31].

In other words, COVID-19 infection is usually accompanied by an exaggerated inflammatory response with the release of a large number of pro-inflammatory cytokines in an event known as “cytokine storm”, which has been directly correlated with lung injury, multi-organ failure, and adverse prognosis [24,25,32]. Few studies provide data on cytokines in children with severe/critical COVID-19 cases, but the most frequently reported elevated cytokines were IL-6 and IFN-γ [32].

Several studies have reported that IL-6 is the cytokine that is most often significantly increased in severe/critical cases compared to mild/moderate cases of COVID-19 in both adults and children [5,24,25,33]. In fact, the optimal results of clinical trials with tocilizumab (an IL-6 inhibitor) in patients with severe and critical COVID-19 led to its approval by the U.S. Food and Drug Administration (FDA) and its inclusion in the current Guidance for the Management of Acute COVID-19 in Children [34].

## 3. Immune Modulatory Activity of Vitamin D

The widespread expression of VDR in immune cells, including antigen-presenting cells (such as dendritic cells and macrophages), as well as T cells and B cells, suggests that vitamin D plays a critical role in the regulation of both the innate and adaptive immune systems. Therefore, vitamin D deficiency may compromise the integrity of the immune system and lead to inappropriate immune responses [35,36,37,38].

Vitamin D is an important regulator of innate immune responses. For example, by modulating the gene expression of potent antimicrobial peptides such as cathelicidin and β-defensin, vitamin D enhances the antimicrobial properties of immune cells (monocytes, neutrophils and natural killer cells) and barrier epithelia, which could reduce viral replication rates. Furthermore, vitamin D has important effects on immune cells, increasing their production of anti-inflammatory cytokines: IL-4 and IL-10, and inhibiting their production of inflammatory cytokines: IL-1, IL-2, IL-6, IL-8, IL-12 and IFN-γ. In this way, it helps prevent an excessive innate response and the resulting tissue damage (systemic inflammation and/or septic shock).

Vitamin D also plays a role in the regulation of adaptive immunity. Vitamin D modulates the activation and differentiation of naïve CD4+ lymphocytes in the lymph nodes, resulting in a shift from a T-helper 1 (Th1) to a Th2 phenotype. This change implies the inhibition of pro-inflammatory cytokine production: IL-2, IFN-γ, and TNF-α, and an increased production of anti-inflammatory cytokines (IL-4, IL-5, and IL-10). Similarly, vitamin D affects differentiation to the Th17 phenotype, leading to a decrease in the production of inflammatory cytokines (IL-17 and IL-21), and facilitates the induction of T regulatory cells (Tregs) with increased production of anti-inflammatory cytokines: IL-10 and transforming growth factor beta (TGF-β) [39,40,41].

In other words, in COVID-19, a link between vitamin D deficiency and the cytokine storm could be established in view of the anti-inflammatory effect of vitamin D. In fact, several studies have shown that vitamin D can reduce the risk and/or the gravity of certain inflammatory reactions [42]. In severe COVID-19, high levels of pro-inflammatory cytokines have been observed, such as IL-1, IL-6, IL-8, IL-12, IL-17, TNF-α and IFN-γ. It appears that vitamin D, through its immunomodulatory role in both innate and adaptive immunity, could potentially prevent or reduce complications associated with the cytokine storm caused by SARS-CoV-2 infection by increasing the levels of anti-inflammatory cytokines (IL-4, IL-5, IL-10), as well as by reducing the levels of pro-inflammatory cytokines [43,44]. In this context, it would be useful to measure the basal level of vitamin D as soon as a diagnosis of SARS-CoV-2 infection has been made. Sequential measurements should also be made during the course of the disease. Whether vitamin D supplementation is associated with a potentially favorable outcome of COVID-19 remains to be seen.

In short, immunological dysregulation would be the main consequence of hypovitaminosis D. For example, patients with SARS-CoV-2 infection have a tendency to develop a hyperinflammatory state and even an overactive pathological immune response. Uncorrected vitamin D deficiency can lead to a “cytokine storm”, increasing the risk of systemic complications (ARDS, multi-organ failure, etc.) and death [45]. In addition, some authors have shown an association between serum vitamin D levels and lymphocyte count in children with COVID-19. This suggests that serum vitamin D concentration may play a role in lymphocyte count, which is very important for the immune system [20].

Vitamin D appears to exert direct and indirect regulation of MMP expression [46] and could therefore act as an inhibitor of MMP proliferation. In fact, vitamin D has been attributed great potential as a therapeutic agent for various human diseases (liver fibrosis, pulmonary emphysema, bone and cartilage diseases, cancer, etc.) by altering the expression and activity of MMPs [47,48].

## 4. Effect of Pandemic Confinement on Vitamin D Status

Vitamin D is mainly produced by the synthesis of the pro-vitamin dehydrocholesterol in the skin under the influence of ultraviolet radiation (type B). The resulting vitamin D is biologically inert and requires double hepatic and renal hydroxylation to become active, also known as “calcitriol” [28,29]. Inadequate exposure to sunlight is therefore one of the main causes of vitamin D deficiency.

Following the rapid spread of COVID-19 and the recommendations of the WHO, many countries adopted restrictive measures to contain the spread of the virus (total closure of schools, universities, public places and all shops except supermarkets, grocery stores and pharmacies). In other words, a series of social distancing measures were implemented. Travel and movement were restricted, and citizens were asked to stay at home to reduce transmission. In children, for example, confinement meant replacing face-to-face schooling with online education and reducing time spent outdoors. In other words, the COVID-19 pandemic forced confinement in all age groups, leading to a significant reduction in time exposed to sunlight and consequently an increased risk of vitamin D deficiency.

Epidemiological studies conducted in China [49,50], Poland [51], Türkiye [52] and Italy [53] have reported that the COVID-19 pandemic-related confinement changed vitamin D status in pediatric-aged patients.

The first observational study to compare vitamin D status in children before and after pandemic-related confinement was conducted in Guangzhou, China [49]. Their results showed an increase in the proportion of children aged 3–6 years with vitamin D deficiency, which was associated with reduced sunlight exposure. However, another study conducted in Hong Kong found evidence of a progressive decline in serum vitamin D levels in infants and young children aged 2–24 months over the course of social distancing and home confinement during the COVID-19 pandemic [50].

In Warsaw, Poland, mean vitamin D levels in the pediatric population aged 1 month to 18 years were reported to be significantly lower during the COVID-19 pandemic (March 2020 to February 2021) than before the pandemic (January 2019 to February 2020), resulting in an increase in the proportion of children with vitamin D deficiency [51]. In addition, the characteristic pre-pandemic seasonal variability of serum vitamin D levels in relation to changes in sunlight exposure was not observed during the pandemic period. Similar results were also obtained in Izmir (Türkiye). That is, vitamin D levels in schoolchildren (6–12 years) and adolescents (12–18 years) were significantly lower in the pandemic period (April 2019–March 2020) than in the pre-pandemic period (April 2020–April 2021) [52]. Likewise, in Naples, Italy, the incidence of hypovitaminosis D was generally higher during the pandemic period (March 2020 to March 2021) than in the pre-pandemic period (March 2019 to March 2020), with higher incidence in preschoolers (1–5 years) and schoolchildren (6–12 years). In this study, no significant differences in serum vitamin D levels were observed in infants (0–12 months) between the pre- and post-COVID periods [53]. Finally, it should be noted that a recent systematic review and meta-analysis study [54] of a large cohort of children and adolescents found significant decreases in vitamin D levels during the pandemic in all pediatric age groups (<18 years) compared with pre-pandemic levels. However, severe vitamin D deficiency was only observed in preschoolers and schoolchildren, and this condition was not observed in infants (less than 1 year of age), probably because of the recommended vitamin D supplementation in this age group [55].

In contrast, in other studies of late adolescents (aged 18–19 years) living in southern Switzerland [56] or children and adolescents (aged 0–18 years) living on the island of Sardinia, Italy [57], the authors reported that vitamin D concentrations remained in the sufficient range during the pandemic period and did not differ significantly from those observed in the pre-pandemic period. Note that these differences could be due to geographical and/or cultural heterogeneity, national differences in severity/duration of confinement, variability in study periods, pediatrician vitamin D supplementation, intake of vitamin D-fortified foods, etc.

## 5. Clinical Manifestations of COVID-19 in Children

The incubation period of SARS-CoV-2, which causes COVID-19, is 3 to 7 days. Infectious sources are predominantly respiratory secretions, although SARS-CoV-2 RNA can be isolated from saliva, blood, feces, and urine. A number of systematic reviews and meta-analyses have described the demographic characteristics, clinical symptoms, laboratory findings and radiological features of COVID-19 [7,18,19,21,22,58]. Based on these data, the following criteria for the severity of COVID-19 infection in children were established [55]: asymptomatic and mild, moderate, severe or critical infection.

The prevalence of asymptomatic infection was 14.2% of children (no clinical symptoms and signs, normal chest imaging, but SARS-CoV-2 infection detected by positive nucleic acid test).

Mild infection was present in 36.3% of children (symptoms of acute upper respiratory tract infection such as fever, fatigue, myalgia, cough, sore throat, runny nose and sneezing; some children may have only gastrointestinal symptoms such as diarrhea, vomiting and abdominal pain). There are no auscultatory abnormalities on physical examination.

Moderate infection (features of viral bronchitis and pneumonia such as fever, productive cough, wheezing, or wet crackles without tachypnea and hypoxemia) occurred in 46% of children. In some cases, there may be no clinical symptoms and signs, but a chest computed tomography scan may reveal subclinical pulmonary lesions.

Severe infections occurred in 2.1% of children (the disease usually progresses in about 1 week and dyspnea occurs with central cyanosis and oxygen saturation less than 92%).

Finally, 1.2% of cases had a critical infection (children can rapidly progress to acute respiratory distress syndrome or respiratory failure). Multiple organ dysfunction such as circulatory shock, encephalopathy, heart failure, disseminated coagulopathy and acute renal failure can be life-threatening).

Figure 1 is a summary of the severity of disease in children with COVID-19. This means that about 95% of the children diagnosed with COVID-19 had an asymptomatic, mild or moderate disease and had a good prognosis with recovery within 1 to 2 weeks. However, the first retrospective study on the epidemiological characteristics of COVID-19 in children was reported by the Chinese CDC in March 2020. This study showed that young children, especially infants, are more susceptible to COVID-19 infection. In fact, the proportion of severe and critical disease was 10.6% in children aged <1 year at diagnosis, 1–5 years (7.3%), 6–10 years (4.2%), 11–15 years (4.1%), and 16–17 years (3.0%) [22].

The most common hematological abnormality in patients with mild COVID-19 was a decreased neutrophil or lymphocyte count, according to laboratory findings reported in several meta-analyses. Biochemical abnormalities were characterized by an increase in the levels of creatine kinase-MB, lactate dehydrogenase (LDH) and liver enzymes: aspartate aminotransferase and alanine aminotransferase. In addition, increased levels of procalcitonin (PCT), D-dimer and C-reactive protein (CRP) were found among laboratory abnormalities related to inflammation/coagulation. Elevated creatine kinase-MB levels may be an indication of cardiac damage, and monitoring of cardiac troponin levels is recommended in hospitalized patients. The higher frequency of elevated PCT levels compared with CRP levels suggests the possibility of viral and bacterial co-infection [18,19,58].

As mentioned above, less than 5% of children with COVID-19 are at risk of developing severe disease; therefore, accumulated data on children with severe COVID-19 are limited. However, leukocyte indices do not appear to be reliable indicators of disease severity in children, according to a pooled analysis and review. In addition, children with severe COVID-19 tended to have elevated levels of LDH, CRP, PCT, D-dimer and serum ferritin, similar to what has been reported in adult patients with COVID-19. Few studies provide data on cytokines in children with severe/critical COVID-19. However, the most commonly reported elevated cytokines were IL-6, IL-8, TNF-α and IFN-γ [58,59,60,61,62].

Chest computed tomography findings of COVID-19 in pediatric patients vary according to the severity of the disease. A recent meta-analysis showed that 41% of cases had normal imaging. The most common lung abnormality was ground-glass opacities (36%) and, to a lesser extent, local (26%) or bilateral (28%) patchy opacities [16,18].

## 6. Multisystem Inflammatory Syndrome

Initial reports indicated that children had mild or asymptomatic COVID-19 and were hospitalized and died at lower rates than adults did [16,21,22,63]. However, within weeks of the COVID-19 pandemic declaration, clinicians in Western Europe and the United States reported clusters of previously healthy children presenting with cardiovascular shock, fever, and Kawasaki disease-like features in the setting of recent SARS-CoV-2 infection or exposure [13,64,65]. This condition has been referred to as the “multi-system inflammatory syndrome temporally associated with COVID-19” (MIS-C). According to the Centers for Disease Control and Prevention (CDC) and WHO, the initial definition of MIS-C included persistent fever (lasting more than 4 days), laboratory data showing inflammation and multiple organ dysfunction associated with recent SARS-CoV-2 infection or exposure within the previous 2–6 weeks, and no other likely explanation. The majority of MIS-C patients were school-aged or adolescents (aged 5–13 years), and available data suggest that the incidence of MIS-C was approximately 3 per 10,000 persons aged <21 years infected with SARS-CoV-2. This means that MIS-C is likely to be a post-infectious inflammatory process that is temporally related to COVID-19 [13,14,15,64,66,67,68].

According to published descriptive studies and systematic reviews covering the largest cohorts of MIS-C patients to date, it is characterized by persistent fever and multi-organ dysfunction (at least four organ systems are involved), including cardiovascular (hypotension, myocardial dysfunction, pericarditis, coronary artery dilation or aneurysm, myocarditis, etc.), mucocutaneous (skin rash, conjunctival injection), respiratory (pneumonia, acute respiratory distress syndrome, pleural effusion, etc.) and gastrointestinal system (abdominal pain, vomiting or diarrhea). It also includes, to a lesser extent, neurologic (headache, meningism, encephalopathy, etc.), nephrological (renal failure) and hepatological (hepatitis or hepatomegaly) symptoms, and a history of previous asymptomatic or mild COVID-19 or exposure to a person with confirmed COVID-19. Their laboratory tests revealed hematology abnormalities (neutrophilia, lymphopenia, and thrombocytopenia), as well as significant increases in inflammation/coagulation markers (erythrocyte sedimentation rate, CRP, PCT, ferritin, fibrinogen, and D-dimer) and elevated cardiac troponin levels [16,66,67,68,69]. Furthermore, these patients had a “cytokine storm” characterized by increased IL-1, IL-6, IL-8, IL-17, TNFα, and IFN-γ [44,70,71]. Although the pathophysiology of MIS-C is still unknown [70], it is thought to be an autoimmune vasculitis in which high levels of anti-SARS-CoV-2 IgG antibodies are associated with a pro-inflammatory cytokine storm. This leads to diffuse endothelial damage by immune complexes and activation of complement pathways [3,71].

More than 80% of MIS-C patients required intensive care for respiratory support and/or hemodynamic instability but recovered with no long-term sequelae and a mortality rate of 1–2% [15,64,67,72,73]. There appears to be a close association between MIS-C and SARS-CoV-2 infection, and MIS-C would therefore be considered a post-infectious manifestation, probably due to an abnormal immune response in most cases.

It is worth noting that previous research has shown elevated baseline levels of some MMPs in children with MIS-C requiring intensive care compared to those with less severe presentations. This means that MMPs may play a role in the pathogenesis of MIS-C and COVID-19 in children and may even serve to differentiate MIS-C from other conditions with overlapping clinical presentations [74,75].

## 7. Clinical Evidence Between COVID-19 and Vitamin D

Regression models showed that the northernmost countries in the Northern Hemisphere had relatively high COVID-19 mortality (May 2020). That is, there appears to be an association between COVID-19 deaths and countries based on latitude (countries with latitudes below 35° north had low mortality rates). This suggests a possible role for vitamin D in determining COVID-19 outcomes, as people living in regions with latitudes above 35° N would receive less exposure to ultraviolet B radiation (an important source of cutaneous vitamin D synthesis) [76,77]. On the other hand, epidemiologic studies have reported that vitamin D deficiency or insufficiency increases susceptibility to infections, such as acute respiratory infections, and that vitamin D supplementation reduces them [45,78]. There is little evidence that vitamin D can protect against SARS-CoV-2 infection through antiviral activity alone, but it may be very important in preventing the cytokine storm and subsequent acute respiratory distress syndrome or MIS-C, which is often the cause of death [29,40,79].

Based on these premises, several studies were conducted to investigate the clinical significance of vitamin D deficiency in both adults and children with COVID-19. According to a systematic review and meta-analysis conducted in adults, including studies from different European (United Kingdom, Italy and Spain) and Asian (Iran, China, Republic of Korea and Saudi Arabia) countries, vitamin D deficiency was identified as an independent risk factor for COVID-19 (patients with vitamin D deficiency are more susceptible to SARS-CoV-2 infection). In addition, vitamin D deficiency was associated with more severe lung injury, acute respiratory distress and risk of death in elderly COVID-19 patients, probably in relation to an immunological imbalance and hyper-inflammatory state [80].

Few studies, mostly observational, have evaluated the relationship between vitamin D levels and clinical severity and inflammatory markers in pediatric SARS-CoV-2 patients because of the milder clinical course of COVID-19. However, several studies have reported similar results in the pediatric population despite differences in methods and statistical approaches. Most of them indicated that vitamin D status seems to have an impact on the outcome of COVID-19 also in children, as low vitamin D levels were associated with a worse prognosis. In fact, serum vitamin D levels were lower in both MIS-C and COVID-19 patients compared to control groups (healthy children). Furthermore, some authors concluded that vitamin D levels could be valuable in the prediction of severe forms of MIS-C and that correction of its abnormal levels in severe MIS-C could have an influence on its evolution [38,81,82,83,84,85,86].

These findings were supported by a meta-analysis of pediatric patients with COVID-19, which included studies from Germany, Romania, the United States, Italy, and Türkiye. Indeed, vitamin D-deficient children and adolescents had a higher risk of contracting SARS-CoV-2 and a worse prognosis compared to those with normal vitamin D levels [87]. Additionally, multivariable logistic regression analyses found that vitamin D deficiency and D-dimer and fibrinogen levels are independent predictive biomarkers of moderate-to-severe clinical outcomes [88,89].

## 8. Why Is COVID-19 Less Severe in Children?

The majority of children infected with COVID-19 are asymptomatic or present with mild or moderate clinical manifestations; namely, symptoms of an acute upper respiratory tract infection or only gastrointestinal symptoms. In the cases with pneumonia, they presented with fever and a dry or productive cough but did not have obvious hypoxemia or respiratory distress [12]. From the early stages of the pandemic, it was observed that children had mild or asymptomatic COVID-19 and had lower rates of hospitalization and death than adults. In fact, since the beginning of the pandemic, age has been identified as an independent predictor of mortality in patients with COVID-19 [16,17,18,19,20,21,22,23,63,90].

Several hypotheses (Table 1) have been proposed to explain the difference in the severity of COVID-19 between children and adults. These hypotheses are based on: (a) factors that would increase the risk in adults, (b) factors that would protect children, or (c) vitamin D status. In reality, these factors are a reflection of the differences in physiological background between children and adults with their clinical consequences [91,92,93].

*(a)* 
*Factors increasing risk in adults*


Viral entry into cells. Angiotensin-converting enzyme 2 (ACE2) receptors are functional receptors of SARS-CoV-2 that are predominantly expressed in the respiratory epithelium (nasopharynx, oropharynx, pneumocytes, etc.) and facilitate viral entry into cells. The expression of the ACE2 gene in the airway epithelium and the affinity of ACE2 receptors for SARS-CoV-2 increases with age and is lower in children than in adults, which would reduce viral entry in children and, therefore, result in a milder infection.

Endothelial damage and hypercoagulable state. Endothelial damage and an excessive susceptibility to clotting increase with age. SARS-CoV-2 can infect endothelial cells and cause vasculitis, resulting in the activation of coagulation pathways and thrombotic complications such as myocardial infarction and stroke in COVID-19. The endothelium in children is “pre-damaged” to a lesser extent than in adults, and the coagulation system is also different, making children less prone to abnormal clotting.

Comorbidities. Children have a lower prevalence of comorbidities associated with severe COVID-19 in adults (obesity, diabetes mellitus, hypertension, immunosuppression, chronic kidney disease, chronic obstructive pulmonary disease, cardiovascular disease, etc.), probably related to endothelial damage.

Intensity of viral exposure. Adults are at higher risk of exposure to SARS-CoV2 due to their work responsibilities (workplace) and social relationships (shopping, travel, etc.), and viral load influences the severity of COVID-19. Therefore, a lower intensity of viral exposure may be another factor favoring less severe disease in children. In addition, since children are usually infected by an adult, they are infected with a second- or third-generation SARS-CoV-2, and subsequent generations of the virus have reduced pathogenicity in comparison to the first-generation virus.

Immunosenescence. Aging is associated with a gradual decline in the function of the innate and adaptive immune systems, which is likely to contribute to the reduced clearance of SARS-CoV-2 [90]. That is, age-related functional defects in the immune system could result in a failure to control viral replication, which could lead to poor outcomes.

*(b)* 
*Factors protecting children*


“Trained” innate immunity. Children infected with SARS-CoV-2 often have co-infections with other viruses (including less pathogenic coronaviruses). Frequent recurrent and concurrent viral infections may interfere with the replication of SARS-CoV-2 by inducing “trained immunity” through epigenetic changes and metabolic reprogramming in innate immune cells, making them more effective in the clearance of SARS-CoV-2.

Off-target effects of live vaccines. Many live vaccines have off-target (non-specific) immunomodulatory effects beyond protection against their target disease. The inclusion of live attenuated vaccines, such as measles, mumps and rubella vaccines and oral polio vaccine, in the childhood immunization schedule would protect against other viral infections by inducing “trained immunity” and/or activating heterologous lymphocytes. This could contribute to differences in the severity of COVID-19.

Age-related differences in immune responses. Children have a stronger innate immune response to cytoplasmic viral pathogens (e.g., SARS-CoV-2) than adults do, with higher natural killer T cell numbers. There is an early type I interferon response (during the incubation period of SARS-CoV-2) and consequently a more rapid activation of host defenses that would lead to more effective containment/elimination of the virus. In terms of adaptive immunity, children also have a higher proportion of T helper 1 lymphocytes (CD4+ T cells), cytotoxic lymphocytes (CD8+ T cells) and activated B lymphocytes than adults, which are essential against viral infections. In addition, it appears that children are less able to generate the pro-inflammatory cytokines storm (lower pro-inflammatory cytokine response); these play an important role in the pathogenesis of severe COVID-19.

*(c)* 
*Vitamin D status*


It is noteworthy that vitamin D is an important regulator (genomic pathway) of both the innate and adaptive immune responses and could therefore play a critical role in the host immune response to SARS-CoV-2 infection. As mentioned above, the mechanism by which vitamin D would play a protective role in COVID-19 would be due to its antiviral activity and, in particular, its anti-inflammatory activity, which would moderate the exaggerated inflammatory response (cytokine storm). Clinical evidence suggests that vitamin D status appears to influence the outcome of COVID-19 in both adults and children, as low vitamin D levels are associated with a worse prognosis.

Finally, it should be noted that MMPs might also be a contributor to the pathophysiological differences between adult and pediatric patients with COVID-19. The role of MMPs in COVID-19 stems from the pathogenesis leading to the release of chemokines and pro-inflammatory markers that cause acute lung injury/acute respiratory distress syndrome. Notably, significantly higher serum concentrations of MMP-3, MMP-8 and MMP-9 have been found in adults with COVID-19 than in pediatric patients with the same condition. This means that adult and pediatric patients with COVID-19 appear to have different MMP responses. Although the specific role of each MMP in COVID-19 is not well understood, it could be hypothesized that several MMPs would not be upregulated in children due to a milder course of infection compared to the adult group [75].

## 9. Vitamin D Supplementation in COVID-19

Indeed, low vitamin D is a recognized risk factor for COVID-19 [79,88,94]. Considering that a severe course of COVID-19 is characterized by an overproduction of pro-inflammatory cytokines (“cytokine storm”) and that vitamin D has an anti-inflammatory effect and also protects against acute respiratory infections in general [45,78], several clinical trials and observational studies have been conducted to evaluate the efficacy of vitamin D supplementation in improving the clinical conditions of patients with COVID-19.

Randomized controlled trials have been conducted in adult patients and have shown that high-dose oral cholecalciferol (60,000 IU daily for 7 days or a single dose of 600,000 IU) is safe and reduces inflammatory biomarkers, sequential organ failure, and mortality in vitamin D-deficient COVID-19 patients [95,96]. In addition, a multicenter randomized clinical trial conducted in Saudi Arabia showed that oral supplementation with 5000 IU of cholecalciferol daily for 2 weeks reduced cough recovery time and taste sensory loss in patients with suboptimal vitamin D status and mild-to-moderate COVID-19 symptoms, and this dose is recommended as adjuvant therapy for COVID-19 patients [97].

The most recent reviews and meta-analyses of randomized clinical trials suggest that cholecalciferol supplementation may reduce hospital length of stay and intensive care unit admission rates in patients infected with SARS-CoV-2. However, the level of evidence is relatively low [3,98,99,100]. This is most likely due to the high heterogeneity of the studies analyzed (different study designs, differences in treatment protocols and diagnosis of COVID-19 infection, patients with different severity of disease, vitamin D supplementation in different forms and dosages, etc.). That is, despite some positive findings, their results should be interpreted with caution (the evidence to support the use of vitamin D supplementation as an adjuvant treatment for SARS-CoV-2 infection is still uncertain). Large-scale randomized controlled trials are needed to confirm the potential benefits of vitamin supplementation in both the prevention and treatment of COVID-19.

At present, the available literature on the efficacy and safety of vitamin D supplementation in pediatric patients with COVID-19 is virtually nonexistent. A recent open-label, randomized, controlled, single-blind clinical trial evaluated the efficacy and safety of vitamin D supplementation (intervention: 2000 IU/day in children aged 1 to 17 years for 7–14 days versus control) in children with COVID-19 requiring hospitalization and supplemental oxygen, without excluding patients with comorbidities (obesity, cancer, and chronic diseases). None of the patients receiving vitamin D had adverse effects. In children with vitamin D supplementation, a decrease in baseline CRP was observed (there was no difference in D-dimer and fibrinogen) and seemed to be able to reduce the risk of COVID-19 clinical deterioration and death. The trial was stopped for ethical reasons since none of the patients in either group had normal vitamin D status after receiving the results of the baseline vitamin D status [101]. That is, despite some positive findings, their results should be interpreted with caution because the evidence supporting vitamin D supplementation as an adjuvant treatment for SARS-CoV-2 infection is still uncertain.

## 10. Conclusions

The clinical manifestations of COVID-19 are highly heterogeneous, ranging from asymptomatic disease to life-threatening conditions (interstitial pneumonia, ARDS or severe multi-organ failure). The normal response to a viral infection requires activation of the inflammatory pathways of the immune system, but COVID-19 infection can generally be associated with an exaggerated release of pro-inflammatory cytokines (cytokine storm), which has been directly correlated with lung injury, multi-organ failure and adverse prognosis. However, some children may experience a post-infectious inflammatory process (MIS-C) associated with COVID-19, probably due to an abnormal immune response. Most of these patients require intensive care for respiratory support and/or hemodynamic instability, but they recover without sequelae and with minimal mortality.

From the early stages of the pandemic, it was observed that most children diagnosed with COVID-19 had asymptomatic, mild or moderate disease and a good prognosis, while older patients infected with SARS-CoV-2 were at high risk of complications and death. Various hypotheses have been proposed for the age-related difference in the severity of COVID-19, including both factors that would increase the risk in adults and factors that would protect children, but there is unlikely to be a single explanation. However, it appears that the immune systems of children are better adapted to clear SARS-CoV-2, and indeed, most children ultimately have mild or asymptomatic disease.

Vitamin D, through its immunomodulatory role in both innate and adaptive immunity, could potentially prevent or reduce complications associated with the cytokine storm caused by SARS-CoV-2 infection by reducing levels of pro-inflammatory cytokines and increasing levels of anti-inflammatory cytokines. Available data would support a role for vitamin D in reducing complications associated with uncontrolled inflammation or the cytokine storm induced by COVID-19. Despite limited clinical data, vitamin D status also appears to affect COVID-19 outcomes in children, as low vitamin D levels have been associated with a worse prognosis.

Although there is no strong evidence to support systematic vitamin D supplementation in patients with SARS-CoV-2 infection, accumulated experience from clinical trials and observational studies suggests that vitamin D supplementation may play a helpful role in the evolution of COVID-19. At present, the available literature on the efficacy and safety of vitamin D supplementation in pediatric COVID-19 patients is practically non-existent. Large-scale randomized controlled trials are needed to confirm the potential benefits of vitamin supplementation in both the prevention and treatment of COVID-19.

## Figures and Tables

**Figure 1 ijms-25-12205-f001:**
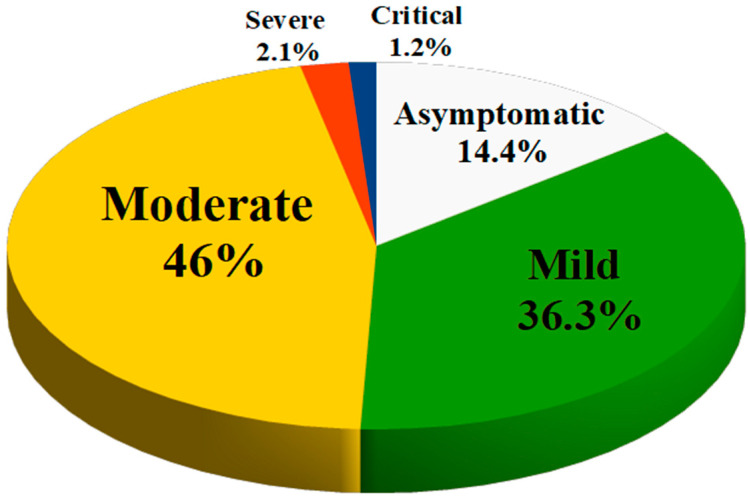
Severity of illness in children with COVID-19.

**Table 1 ijms-25-12205-t001:** Differences in the physiological responses of children and adults to SARS-CoV-2.

**Factors That Increase the Risk of COVID-19 in Adults**	
Receptors and their affinity to SARS-CoV-2	Higher expression of ACE2 in the respiratory tract and higher affinity to SARS-CoV-2 in adults than in children.
Endothelial damage and hypercoagulable state	With age, there is an increase in the damage to the endothelium and an excessive susceptibility to blood clotting. SARS-CoV-2 can cause vasculitis by activating coagulation pathways (thrombotic complications).
Associated comorbidities	Higher prevalence of risk-related comorbidities (obesity, diabetes mellitus, hypertension, immunosuppression, chronic diseases, etc.) in adults.
Immunosenescence	Aging is associated with a gradual decline in immune function, which probably contributes to reduced SARS-CoV-2 clearance.
Intensity of viral exposure	Adults are at higher risk of exposure to SARS-CoV2 given their work responsibilities and social relationships, and viral load influences the severity of COVID-19.
**Factors That Protect Against COVID-19 in Children**	
“Trained” innate immunity	Frequent recurrent and concurrent viral infections in children could interfere with SARS-CoV-2 replication by inducing “trained immunity”.
Off-target effects of live vaccines	Live attenuated vaccines have off-target (non-specific) immunomodulatory effects beyond protection against their target disease by inducing “trained immunity” or heterologous lymphocyte activation.
Age-related differences in immune response	Children have a stronger and faster innate immune response (during the incubation period of SARS-CoV-2) than adults. Children have a higher proportion of lymphocytes (T and B cells) than adults. Children have less capacity to generate the pro-inflammatory cytokine storm than adults.
**Vitamin D Status**	
Vitamin D, through its immunomodulatory function of the immune system (anti-inflammatory agent), might potentially moderate the cytokine storm, which plays an important role in the pathogenesis of severe COVID-19.	

## Data Availability

The datasets generated during and/or analyzed during the current study are available from the corresponding author upon reasonable request.

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
