# Peer review of "COVID-19 in Children and Vitamin D"

_ijms, 2024, doi:10.3390/ijms252212205_

Round 1
Reviewer 1 Report
Comments and Suggestions for Authors
This is an important topic and this review can potentially contribute to the understanding of the impact of vitamin D to reduce morbidity of COVID-19 in children (although results from clinical treatments did not show effects so far).
I have to points in the text and Figure, and a suggestion.
In line 240, it was difficult (at least for me) in the beginning to understand that those categories of severity stem from reference 52, which is cited just in the paragraph above. I think, maybe the authors could say this explicitly in the sentence... such as "according to the criteria ... (ref), the grades of clinical presentation are ... (or something like that).
In Figure 1, again, a note on the reference (52, probably) with year of publication. In my view, this needs to appear, otherwise the reader does not know from where this pizza graph was taken.
In the Table "Factors that increase the risk OF COVID-19 in adults" (please complete the title in the table title), in my, view, one important risk factor is the amount of MMP.
All chronic diseases presented AND aging are known to be associated with increased expression and activity of MMPs, particularly MMP-2, MMP-8, and MMP-9. MMP-9 is a risk factor for cardiovascular disease sometimes considered as important as Reactive Protein C.
There is plenty of (indirect) data showing that periodontal patients (which have increased circulating MMPs, as well as more MMPs destroying the bone) have a higher susceptibility of die from COVID-19. This is also so with cardiovacular patientes and metabolic disease patients, such as diabetic and metabolic syndrome patients.
And there are papers showing that the citokine storm is associated with increased MMPs in plasma and in the lungs of severe COVID-19 adult patients. doi: 10.3390/biom12050604.
The precise mechanism by which Vitamin D regulates expression and activity of MMPs is now known, but there is a large amount of studies showing that vitamin D decreases MMP activities, both in cardiac diseases, renal diseases DOI: 10.5414/CN109121), in the skin (in diabetic skin wounds), in bone and cartilage diseases, and interestingly in the liver, vitamin D decreases the progression of liver from steatosis to fibrosis. So, it might at least be important to cite this interesting link. It is so well sedimented in the literature that the imbalance of MMP/TIMPs increases over the ages.
Congratulations for the interesting work.
Author Response
Reply to reviewer-1
Comments and Suggestions for Authors
First of all, we would like to thank you for your suggestions as well as your words of encouragement regarding this article.
NOTE: The corrected text of the new version is in red
This is an important topic and this review can potentially contribute to the understanding of the impact of vitamin D to reduce morbidity of COVID-19 in children (although results from clinical treatments did not show effects so far).
I have to points in the text and Figure, and a suggestion.
(1) In line 240, it was difficult (at least for me) in the beginning to understand that those categories of severity stem from reference 52, which is cited just in the paragraph above. I think, maybe the authors could say this explicitly in the sentence... such as "according to the criteria ... (ref), the grades of clinical presentation are ... (or something like that).
In section 5 (Clinical manifestations of COVID-19 in children)
Previous text...
Based on these data, criteria for the severity of COVID-19 infection in children were established, including the following clinical types: asymptomatic infection and mild, moderate, severe or critical infection [52].
…has been changed to... (lines: 234-236)
“Based on these data, the following criteria for the severity of COVID-19 infection in children were established (52): asymptomatic and mild, moderate, severe or critical infection.”
(2) In Figure 1, again, a note on the reference (52, probably) with year of publication. In my view, this needs to appear, otherwise the reader does not know from where this pizza graph was taken.
The legend of the figure 1 has been enlarged for a better understanding
Previous text...
Figure 1. Severity of illness in children with COVID-19
…has been changed to...
Figure 1. Severity of illness in children with COVID-19 (adapted from 52).
(3) In the Table "Factors that increase the risk OF COVID-19 in adults" (please complete the title in the table title), in my, view, one important risk factor is the amount of MMP.
In table 1:
Previous text...
Factors that increase the risk in adults.
Factors that protect children.
…has been changed to…
Factors that increase the risk of COVID-19 in adults.
Factors that protect against COVID-19 in children.
The potential importance of MMPs as a factor that increases the risk in adults will be discussed later.
(4) All chronic diseases presented AND aging are known to be associated with increased expression and activity of MMPs, particularly MMP-2, MMP-8, and MMP-9. MMP-9 is a risk factor for cardiovascular disease sometimes considered as important as Reactive Protein C. There is plenty of (indirect) data showing that periodontal patients (which have increased circulating MMPs, as well as more MMPs destroying the bone) have a higher susceptibility of die from COVID-19. This is also so with cardiovacular patientes and metabolic disease patients, such as diabetic and metabolic syndrome patients. And there are papers showing that the citokine storm is associated with increased MMPs in plasma and in the lungs of severe COVID-19 adult patients. doi: 10.3390/biom12050604. The precise mechanism by which Vitamin D regulates expression and activity of MMPs is now known, but there is a large amount of studies showing that vitamin D decreases MMP activities, both in cardiac diseases, renal diseases DOI: 10.5414/CN109121), in the skin (in diabetic skin wounds), in bone and cartilage diseases, and interestingly in the liver, vitamin D decreases the progression of liver from steatosis to fibrosis. So, it might at least be important to cite this interesting link. It is so well sedimented in the literature that the imbalance of MMP/TIMPs increases over the ages.
In accordance with your suggestion regarding the role of MMS in covid-19, the following paragraphs/sentences have been added:
Section 1 (Introduction). Lines: 94-97.
I In addition, recent clinical data [25] have shown that patients with COVID-19 have significantly elevated levels of matrix metalloproteinase (MMPs), which are known to play a role in tissue remodeling and immune responses. In other words, MMPs, together with the cytokine storm, may play an important role in the immunopathogenesis of COVID-19.
Section 3 (Immune modulatory activity of vitamin D). Lines: 178-181.
Vitamin D appears to exert direct and indirect regulation of MMP expression [43] and could therefore act as an inhibitor of MMP proliferation. In fact, vitamin D has been attributed great potential as a therapeutic agent for various human diseases (liver fibrosis, pulmonary emphysema, bone and cartilage diseases, cancer, etc.) by altering the expression and activity of MMPs [44, 45].
Section 6 (Multisystem inflammatory syndrome). Lines: 312-315.
It is worth noting that previous research has shown elevated baseline levels of some MMPs in children with MIS-C requiring intensive care compared to those with less severe presentations. This means that MMPs may play a role in the pathogenesis of MIS-C and COVID-19 in children and may even serve to differentiate MIS-C from other conditions with overlapping clinical presentations [71, 72].
Section 8 (Why is COVID19 less severe in children?). Lines: 418-425.
Finally, it should be noted that MMPs might also be a contributor to the pathophysiological differences between adult and paediatric patients with COVID-19. The role of MMPs in COVID-19 stems from the pathogenesis leading to the release of chemokines and pro-inflammatory markers that cause acute lung injury/acute respiratory distress syndrome. Notably, significantly higher serum concentrations of MMP-3, MMP-8 and MMP-9 have been found in adults with COVID-19 than in paediatric patients with the same condition. This means that adult and paediatric patients with COVID-19 appear to have different MMP responses. Although the specific role of each MMP in COVID-19 is not well understood, it could be hypothesised that several MMPs would not be upregulated in children due to a milder course of infection compared to the adult group [72].
We would like to express our thanks to referee for your suggestions and positive criticisms.
We hope every made question have been answered adequately.
Yours sincerely,

Reviewer 2 Report
Comments and Suggestions for Authors
The manuscript titled "COVID-19 in Children and Vitamin D" presents an investigation into the role of vitamin D, showcasing the authors' commitment to addressing an important health issue. The topic is timely and relevant, especially given the ongoing discussions surrounding vitamin D's impact on immune health. However, the manuscript has several important flaws that need to be addressed. While the study aims to provide insights into the relationship between vitamin D and COVID-19 in children, the background information lacks depth, leaving significant gaps in the rationale for the study. Moreover, several questions warrant further clarification.
I have several crucial suggestions for improvement. The authors should consider adding more tables and figures to enhance the clarity of their findings and support their arguments. For instance, a comprehensive summary table of the meta-analysis would give readers a succinct overview of the evidence. Additionally, including more figures illustrating the physiological differences between children and adults in response to COVID-19 could significantly enrich the manuscript. Making these revisions would strengthen the overall quality and impact of the review. My questions and suggestions are attached.

Author Response
Reply to reviewer-2
We sincerely appreciate the comments of this reviewer. However, we think that your comments and suggestions, while exquisitely insightful, are out of place in this case.
The format of this article corresponds to a narrative review, whose elaboration methods and proposed objectives are very different from those of a scoping review and, of course, from those of a systematic review. In fact, the aim of a narrative review (such as this article) is not to exhaust a particular medical problem, but to summarize relevant evidence on a particular topic that might be of interest to other health science professionals. However, despite their limitations, narrative reviews have an important place in the dissemination of biomedical information because they bring readers up to date on particular topics.
To answer all the questions raised by the reviewer would require an encyclopaedic knowledge of the subject. Obviously, as this is a new medical problem, many multidisciplinary issues remain unresolved, but this was not the aim of this narrative review. In fact, as we consulted the various bibliographic references to prepare this article, most of the questions suggested by the reviewer arose, but without a clear answer. And the purpose of any narrative review would be to provide basic but necessary information for clinical practice and to raise concerns for future researchers.
Yours sincerely,

Reviewer 3 Report
Comments and Suggestions for Authors
According to the title, this paper aimed to search for correlations between pediatric SARS-Covid-2 patients and vitamin D level. However, the manuscript itself was not organized well to focus on this topic.
Authors used too much effort describing the general information about COVID-19 infections. In section 4, it seems the conclusion is that COVID-19 quarantine leads to low vitamin D level due to less outdoor activity. Whereas in later paragraphs, low vitamin D level is considered as a risk factor and was associated with a worse prognosis. It's quite confusing that if the authors mean low vitamin D is a causal factor, or the result of COVID-19 infection.
Also, too much content is about general population rather than children/adolescences. Take section 8 as an example, children would be the subject, not the adults, in a review focus on pediatric patients.
I would suggest the authors to update the paper's structure, rearrange contents, delete irrelevant information to improve this narrative review.
Author Response
Reviewer 3
Comments and Suggestions for Authors
According to the title, this paper aimed to search for correlations between pediatric SARS-Covid-2 patients and vitamin D level. However, the manuscript itself was not organized well to focus on this topic.
Authors used too much effort describing the general information about COVID-19 infections. In section 4, it seems the conclusion is that COVID-19 quarantine leads to low vitamin D level due to less outdoor activity. Whereas in later paragraphs, low vitamin D level is considered as a risk factor and was associated with a worse prognosis. It's quite confusing that if the authors mean low vitamin D is a causal factor, or the result of COVID-19 infection.
Also, too much content is about general population rather than children/adolescences. Take section 8 as an example, children would be the subject, not the adults, in a review focus on pediatric patients.
I would suggest the authors to update the paper's structure, rearrange contents, delete irrelevant information to improve this narrative review.
First of all, we would like to thank you for your advice regarding this article.
The idea of doing this narrative review arose when we noticed the abundant bibliography relating COVID-19 to vitamin D status. And, of course, given our status as paediatricians, it would focus on children.
We thought about the different sections we should develop, and we realized that the order we had chosen seemed quite reasonable.
Although the title is very general (as is often the case), the objectives were more specific. Section 1 (Introduction) gives a brief introductory idea of the different themes that will be developed throughout the article. In addition, the objectives of this review are specifically listed.
We felt it was essential to continue with sections 2 (The COVID-19 cytokine storm) and 3 (The immunomodulatory activity of vitamin D). On the one hand, SARS-CoV-2 infection is often associated with the release of large amounts of pro-inflammatory cytokines, a so-called "cytokine storm"; this correlates with a poor prognosis in severe cases of COVID-19. On the other hand, the regulatory role of vitamin D in the innate and adaptive immune response (anti-inflammatory effect) needed to be clarified as soon as possible. In other words, the vitamin D status in COVID-19 could play an important immunomodulatory role by regulating the pathophysiological manifestations of the cytokine storm.
Precisely because of what has been stated in the previous sections (2 and 3), we thought it appropriate to continue with section 4 (The effect of pandemic confinement on vitamin D status). Home confinement was necessary to control the spread of the virus. However, these restrictive measures resulted in a significant reduction in sunlight exposure time and consequently an increased risk of vitamin D deficiency. Epidemiological studies conducted in different countries confirmed a progressive decrease in serum vitamin D levels in children during home confinement due to the COVID-19 pandemic. This means that home confinement had undesirable inconveniences that could have negatively affected the development of SARS-CoV-2 infection. We therefore decided that this topic deserved its own section.
Section 5 (Clinical manifestations of COVID-19 in children) is a description of the grades of clinical presentation of COVID-19 in children. Approximately 95% of the children diagnosed with COVID-19 were found to be asymptomatic, mild or moderate affected, and with a good prognosis. Section 6 (Multisystem inflammatory syndrome) describes a post-infectious inflammatory process temporally related to COVID-19, called "multisystem inflammatory syndrome" (MIS-C). In most cases, it is probably due to an abnormal immune response ("cytokine storm"). Its incidence was about 3 per 10,000 persons aged <21 years infected with SARS-CoV-2, with a mortality rate of 1-2%. These sections (5 and 6) seemed very appropriate to us because they would explain the few studies, mostly observational, that have evaluated the relationship between vitamin D levels and clinical severity and inflammatory markers in paediatric patients with SARS-CoV-2, precisely because of the milder clinical course of COVID-19.
Section 7 (Clinical evidence between COVID-19 and vitamin D) states that "...there is little evidence that vitamin D can protect against SARS-CoV-2 infection by antiviral activity alone, but it may be very important in preventing the cytokine storm and subsequent acute respiratory distress syndrome or MIS-C...". In addition, several studies conducted in different countries on paediatric patients with COVID-19 are mentioned, which support that children and adolescents with vitamin D deficiency had a higher risk of contracting SARS-CoV-2 and a worse prognosis compared to those with normal vitamin D levels. Nowhere does the study suggest that vitamin D deficiency caused or contributed to SARS-CoV infection. As noted in the text, epidemiological studies have reported that vitamin D deficiency or insufficiency increases the susceptibility to infections, such as acute respiratory infections, and that vitamin D supplementation reduces the susceptibility to these infections.
From the early stages of the pandemic, it was observed that children had mild or asymptomatic COVID-19 and lower rates of hospitalization and death than adults. Section 8 (Why is COVID-19 less severe in children?) presents several hypotheses to explain the difference in COVID-19 severity between children and adults. Inevitably, adult patients are mentioned in this section because the proposed hypotheses compare the differences in physiological responses of children and adults to SARS-Cov-2 (Table 1).
Section 9 (Vitamin D supplementation in COVID-19) aims to evaluate the efficacy of vitamin D supplementation in paediatric patients with COVID-19. Most randomised controlled trials have been conducted in adult patients and have shown that vitamin D supplementation is safe and reduces inflammatory biomarkers, hospital length of stay and intensive care unit admission rates in patients infected with SARS-CoV-2. At present, the available literature on the efficacy and safety of vitamin D supplementation in paediatric patients with COVID-19 is virtually non-existent, again due to the milder clinical course of COVID-19. This means that despite some positive findings, their results should be interpreted with caution, as the evidence supporting vitamin D supplementation as an adjuvant treatment for SARS-CoV-2 infection is still uncertain.
Finally, Section 10 (Conclusions) presents the conclusions of this narrative review.
We would like to express our gratitude to the reviewer for his suggestions, but – with all due respect – we are convinced that the structure of the article and the organization of its contents fit the objectives of this narrative review.
Yours sincerely,

Round 2
Reviewer 2 Report
Comments and Suggestions for Authors
The response provided by the authors fails to acknowledge the validity and necessity of the questions and suggestions I raised. While the narrative review format is less exhaustive than a systematic review, it is not an excuse to omit critical information on a topic as impactful and nuanced as COVID-19 in children and the role of vitamin D. My comments aimed to address evident gaps in the manuscript that, if left unaddressed, compromise the scientific rigor and practical utility of the review. A narrative review must still meet high standards of completeness and clarity, especially in discussing complex biomedical topics that carry real implications for clinical practice and public health. Dismissing these suggestions under the guise of format limitations is not acceptable. Fundamental gaps—such as lacking clarity on physiological differences between age groups or on mechanisms by which vitamin D may modulate immune responses—are essential to the relevance and accuracy of this review. Future responses to reviewers should reflect a commitment to constructive revision, rather than bypassing valid critique based on format limitations. In conclusion, the authors' response falls short of addressing the critical feedback necessary to enhance the scientific integrity and relevance of this review. The lack of detailed information on key areas such as age-specific physiological responses, vitamin D's immune mechanisms, and other important questions I raised limits the manuscript's value and applicability. As it stands, I cannot recommend acceptance of this review in its current form, as doing so would compromise the journal's standards for quality and thoroughness in publishing biomedical literature.
Author Response
Dear reviewer,
We found your questions and suggestions to be very eloquent and reflect excellent scientific concern and ambition. However, we understand that there are still many questions to be answered about COVID-19 in children and the role of vitamin D, and that it will be up to the scientific community - in a multidisciplinary way - to respond to the questions raised in a timely manner. However, the introduction of vaccines against SARS-CoV-2 has changed the course of the pandemic and some issues will be difficult to resolve.
In addition, due to the milder clinical course of COVID-19, paediatricians face the difficulty that few studies, mostly observational, have evaluated the relationship between vitamin D levels and clinical severity and inflammatory markers in paediatric patients with SARS-CoV-2. Currently, there is a paucity of literature on the efficacy and safety of vitamin D supplementation in paediatric patients with COVID-19.
However, it is mentioned throughout the text that COVID-19 infection is often accompanied by an aggressive inflammatory response with the release of large amounts of pro-inflammatory cytokines (cytokine storm), which has been directly correlated with lung injury, multi-organ failure and adverse prognosis. The immunomodulatory role of vitamin D in both innate and adaptive immunity (the immune mechanisms of vitamin D) is also revealed, which could potentially prevent or reduce complications associated with the cytokine storm caused by SARS-CoV-2 infection by increasing levels of anti-inflammatory cytokines and reducing levels of pro-inflammatory cytokines.
The age-related differences in immunological responses to SARS-CoV-2, the role of vitamin D in regulating the severity of COVID-19 (model of immunomodulation by vitamin D in COVID-19) and the differences in pathomechanism between children and adults with COVID-19 have been studied by several authors [Kalia et al, 2020 (ref. 40). Wong et al, 2020 (ref. 91). Zimmermann, 2021 (ref. 92). Kapustova et al, 2021 (ref. 93); ]. In this narrative review, we aimed to synthesise the contributions of these authors with the greatest integrity and clarity possible.
We would like to express our gratitude to the reviewer for his comments and suggestions, but – with all due respect – we are convinced that the structure of the article and the organization of its contents are in line with the objectives of this narrative review.
Sincerely,